# OPIEC: An Open Information Extraction Corpus

**Kiril Gashteovski**                                          K.GASHTEOVSKI@UNI-MANNHEIM.DE
**Sebastian Wanner**                                          SWANNER@MAIL.UNI-MANNHEIM.DE
**Sven Hertling**                                            SVEN@INFORMATIK.UNI-MANNHEIM.DE
**Samuel Broscheit**                                    BROSCHEIT@INFORMATIK.UNI-MANNHEIM.DE
**Rainer Gemulla**                                          RGEMULLA@UNI-MANNHEIM.DE
*Universität Mannheim, Germany*

## Abstract

Open information extraction (OIE) systems extract relations and their arguments from natural language text in an unsupervised manner. The resulting extractions are a valuable resource for downstream tasks such as knowledge base construction, open question answering, or event schema induction. In this paper, we release, describe, and analyze an OIE corpus called OPIEC, which was extracted from the text of English Wikipedia. OPIEC complements the available OIE resources: It is the largest OIE corpus publicly available to date (over 340M triples) and contains valuable metadata such as provenance information, confidence scores, linguistic annotations, and semantic annotations including spatial and temporal information. We analyze the OPIEC corpus by comparing its content with knowledge bases such as DBpedia or YAGO, which are also based on Wikipedia. We found that most of the facts between entities present in OPIEC cannot be found in DBpedia and/or YAGO, that OIE facts often differ in the level of specificity compared to knowledge base facts, and that OIE open relations are generally highly polysemous. We believe that the OPIEC corpus is a valuable resource for future research on automated knowledge base construction.

## 1. Introduction

Open information extraction (OIE) is the task of extracting relations and their arguments from natural language text in an unsupervised manner [Banko et al., 2007]. The output of such systems is usually structured in the form of (*subject*, *relation*, *object*)-triples. For example, from the sentence *"Bell is a telecommunication company, which is based in L. A.,"* an OIE system may yield the extractions *("Bell"; "is"; "telecommunication company")* and *("Bell"; "is based in"; "L. A.")*. The extractions of OIE systems from large corpora are a valuable resource for downstream tasks [Etzioni et al., 2008, Mausam, 2016] such as automated knowledge base construction [Riedel et al., 2013, Wu et al., 2018, Vashishth et al., 2018, Shi and Weninger, 2018], open question answering [Fader et al., 2013], event schema induction [Balasubramanian et al., 2013], generating inference rules [Jain and Mausam, 2016], or for improving OIE systems themselves [Mausam et al., 2012, Yahya et al., 2014]. A number of derived resources have been produced from OIE extractions, including as entailment rules [Jain and Mausam, 2016], question paraphrases [Fader et al., 2013], Rel-grams [Balasubramanian et al., 2012], and OIE-based embeddings [Stanovsky et al., 2015].

In this paper, we release a new OIE corpus called *OPIEC*.[1] The OPIEC corpus has been extracted from the full text of the English Wikipedia using the Stanford CoreNLP pipeline [Manning et al., 2014] and the state-of-the-art OIE system MinIE [Gashteovski et al., 2017]. OPIEC complements available OIE resources [Fader et al., 2011, Lin et al., 2012, Nakashole et al., 2012, Moro and Navigli, 2012, 2013, Delli Bovi et al., 2015b,a]: It is the largest OIE corpus publicly available to date (with over 340M triples) and contains valuable metadata information for each of its extractions not available in existing resources (see Tab. 1 for an overview). In particular, OPIEC provides for each triple detailed provenance information, syntactic annotations (such as POS tags, lemmas, dependency parses), semantic annotations (such as polarity, modality, attribution, space, time), entity annotations (NER types and, when available, Wikipedia links), as well as confidence scores.

We performed a detailed data profiling study of the OPIEC corpus to analyze its contents and potential usefulness for downstream applications. We observed that a substantial fraction of the OIE extractions was not self-contained (e.g., because no anaphora resolution was performed) or overly specific (e.g., because arguments were complex phrases). Since these extractions are more difficult to work with, we created the *OPIEC-Clean* subcorpus (104M triples), in which we only retained triples that express relations between concepts. In particular, OPIEC-Clean contains triples in which arguments are either named entities (as recognized by an NER system), match a Wikipedia page title (e.g., concepts such as *political party* or *movie*), or link directly to a Wikipedia page. Although OPIEC-Clean is substantially smaller than the full OPIEC corpus, it is nevertheless four times larger than the largest prior OIE corpus.

To gain insight into the information present in the OPIEC corpus, we compared its content with the DBpedia [Bizer et al., 2009] and YAGO [Hoffart et al., 2013] knowledge bases, which are also constructed from Wikipedia (e.g., from infoboxes). Since such an analysis is difficult to perform due to the openness and ambiguity of OIE extractions, we followed standard practice and used a simple form of distant supervision. In particular, we analyze the *OPIEC-Linked* subcorpus (5.8M triples), which contains only those triples in which both arguments are linked to Wikipedia articles, i.e., where we have golden labels for disambiguation. We found that most of the facts between entities present in OPIEC-Linked cannot be found in DBpedia and/or YAGO, that OIE facts often differ in the level of specificity compared to knowledge base facts, and that frequent OIE open relations are generally highly polysemous.

Along with the OPIEC corpus as well as the OPIEC-Clean and OPIEC-Linked subcorpora, we release the codebase used to construct the corpus as well as a number of derived resources, most notably a corpus of open relations between arguments of various entity types along with their frequencies. We believe that the OPIEC corpus is a valuable resource for future research on automated knowledge base construction.

## 2. Related OIE Corpora

Many structured information resources created from semi-structured or unstructured data have been constructed in recent years. Here we focus on large-scale OIE corpora, which do not make use of a predefined set of arguments and/or relations. OIE corpora complement

---

1. The OPIEC corpus is available at https://www.uni-mannheim.de/dws/research/resources/opiec/.

| | # triples (millions) | # unique arguments (millions) | # unique relations (millions) | disamb. args (aut./gold) | confi-dence | prove-nance | syntactic annotat. | semantic annotat. |
|---|---|---|---|---|---|---|---|---|
| ReVerb | 14.7 | 2.2 | 0.7 | - / - | ✓ | ✓ | ✓ | - |
| ReVerb-Linked | 3.0 | 0.8 | 0.5 | ✓ / - | - | - | - | - |
| PATTY (Wiki) | 15.8 | 0.9 | 1.6 | ✓ / - | - | - | - | - |
| WiseNet 2.0 | 2.3 | 1.4 | 0.2 | - / ✓ | - | - | - | - |
| DefIE | 20.3 | 2.5 | 0.3 | ✓ / - | - | - | - | - |
| KB-Unify | 25.5 | 2.1 | 2.3 | ✓ / - | - | - | - | - |
| OPIEC | 341.0 | 104.9 | 63.9 | - / - | ✓ | ✓ | ✓ | ✓ |
| OPIEC-Clean | 104.0 | 11.1 | 22.8 | - / - | ✓ | ✓ | ✓ | ✓ |
| OPIEC-Linked | 5.8 | 2.1 | 0.9 | - / ✓ | ✓ | ✓ | ✓ | ✓ |

Table 1: Available OIE corpora and their properties. All numbers are in millions. Syntactic annotations include POS tags, lemmas, and dependency parses. Semantic annotations include attribution, polarity, modality, space, and time.

more targeted resources such as knowledge bases (e.g., DBpedia, YAGO) or NELL [Mitchell et al., 2018], as well as smaller, manually crafted corpora such as the one of Stanovsky and Dagan [2016]; see also the discussion in Sec. 5. An overview of the OPIEC corpus, its subcorpora, and related OIE corpora is given in Tab. 1.

One of the first and widely-used OIE resource is the ReVerb corpus [Fader et al., 2011], which consists of the high-confidence extractions of the ReVerb OIE system from the ClueWeb09 corpus. In subsequent work, a subset of the ReVerb corpus with automatically disambiguated arguments was released [Lin et al., 2012]. The PATTY [Nakashole et al., 2012], WiseNet [Moro and Navigli, 2012], WiseNet 2.0 [Moro and Navigli, 2013], and DefIE [Delli Bovi et al., 2015b] corpora additionally organize open relations in relational synsets and then the relational synsets into taxonomies. Finally, KB-Unify [Delli Bovi et al., 2015a] integrates multiple different OIE corpora into a single resource.

The largest prior corpus is KB-Unify, which consists of 25.5M triples in roughly 4.5M distinct open relations and arguments. Both OPIEC (341M and 105M/64M, resp.) and OPIEC-Clean (104M and 11M/23M) are significantly larger than KB-Unify, both in terms of number of triples as well as in terms of distinct arguments and relations. One of the reasons for this size difference is that the MinIE extractor, which we used to create the OPIEC corpus, produces more extractions than the extractors used to create prior resources. The OPIEC corpus—but not OPIEC-Clean and OPIEC-Linked–also contains all extractions produced by MinIE unfiltered, whereas most prior corpora use filtering rules aiming to provide higher-quality extractions (e.g., frequency constraints).

Most of the available corpora use automated methods to disambiguate entities (e.g., w.r.t. to a knowledge base). On the one hand, such links are very useful because ambiguity is restricted to open relations. On the other hand, the use of automated entity linkers may introduce errors and—perhaps more importantly—restricts the corpus to arguments that can be confidently linked. We did not perform automatic disambiguation in OPIEC, although we retained Wikipedia links when present (similar to WiseNet). Since these links are provided by humans, we consider them as golden disambiguation links. The OPIEC-Linked subcorpus contains almost 6M triples from OPIEC in which both arguments are disambiguated via such golden links.

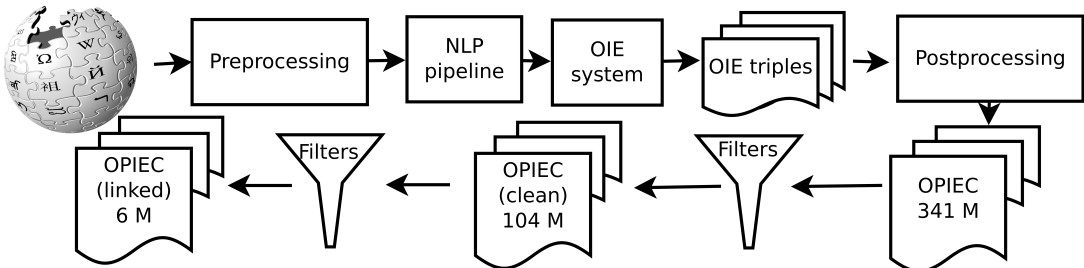

Figure 1: Corpus construction pipeline

A key difference between OPIEC and prior resources is in the amount of metadata provided for each triple. First, only ReVerb and OPIEC provide confidence scores for the extractions. The confidence score measures how likely it is that the triple has been extracted correctly (but not whether it is actually true). For example, given the sentence *"Bill Gates is a founder of the Microsoft."*, the extraction *("Bill Gates"; "is founder of"; "Microsoft")* is correct, whereas the extraction *("Bill Gates"; "is"; "Microsoft")* is not. Since OIE extractors are bound to make extraction errors, the extractor confidence is an important signal for downstream applications [Dong et al., 2014]. Similarly, provenance information (i.e., information about where the triple was extracted) is not provided in many prior resources.

One of the reasons why we chose MinIE to construct OPIEC is that it provides syntactic and semantics annotations for its extractions. Syntactic annotations include part-of-speech tags, lemmas, and dependency parses. Semantic annotations include attribution (source of information according to sentence), polarity (positive or negative), and modality (certainty or possibility). We also extended MinIE to additionally provide spatial and temporal annotations for its triples. The use of semantic annotations simplifies the resulting triples significantly and provides valuable contextual information; see the discussion in Section 3.

## 3. Corpus Construction

OPIEC was constructed from all the articles of the English Wikipedia dump of June 21, 2017. Fig. 1 gives an overview of the pipeline that we used. The pipeline is (apart from preprocessing) not specific to Wikipedia and thus can be used with other dataset as well. We used Apache Spark [Zaharia et al., 2016] to distribute corpus construction across a compute cluster so that large datasets can be handled. We release the entire codebase along with the actual OPIEC corpora.

### 3.1 Preprocessing

We used a modified version of WikiExtractor[2] to extract the plain text from Wikipedia pages. In particular, we modified WikiExtractor such that it retains internal Wikipedia links to other Wikipedia articles. The links are provided as additional metadata in the

---

2. https://github.com/attardi/wikiextractor

form of (*span*, *target page*) annotations. Custom entity linkers can be inserted into the pipeline by providing such annotations.

Wikipedia generally does not link the first phrase of an article to the article page itself; e.g., the page on New Hampshire starts with *"New Hampshire is a ..."*, where *New Hampshire* is not linked. To avoid losing this relationship, we link the first phrase that *exactly* matches the Wikipedia page name (if any) to that Wikipedia page.

## 3.2 NLP Pipeline

We ran an NLP pipeline on the preprocessed Wikipedia articles by using CoreNLP [Manning et al., 2014], version 3.8.0. We performed tokenization, sentence splitting, part-of-speech tagging, lemmatization, named entity recognition (NER) [Finkel et al., 2005], temporal tagging [Chang and Manning, 2012], and dependency parsing [Chen and Manning, 2014].

## 3.3 The MinIE-SpaTe OIE System

From each of the resulting sentences, we extract triples using the state-of-the-art OIE system MinIE [Gashteovski et al., 2017], which in turn in based on ClausIE [Del Corro and Gemulla, 2013]. MinIE *minimizes* the extractions into more compact triples by removing unnecessary words (e.g. determiners) without damaging the semantic content of the triple and by providing *semantic annotations*. The semantic annotations move auxiliary information from the triple (thereby simplifying it) to annotations. The semantic annotations include polarity, modality, attribution, and quantities. Polarity (positive or negative) indicates whether or not the triple occurred in negated form in the input sentence. *Modality* indicates whether the triple is a certainty (CT) or merely a possibility (PS) according to the input. *Attribution* refers to the supplier of the information carried within the triple, again according to the input sentence. Attributions have their own polarity and modality. Finally, quantities express specific amounts.

Consider for example the sentence: *"David Heyman said that Gilderoy Lockhart will probably not be played by Kenneth Branagh."* MinIE extracts the following triple and annotations:

("Gilderoy Lockhart"; "be played by"; "Kenneth Branagh")
*Factuality:* negative possibility
*Attribution:* "David Heyman", (positive certainty)

Many of the sentences in Wikipedia contain some sort of temporal or spatial reference (roughly 56% according to a preliminary study). Since this information is important, we modified MinIE's output to add additional semantic annotations for space and time, thereby producing SPOTL facts [Hoffart et al., 2013]. We refer to the resulting system as MinIE-SpaTe. Generally, MinIE-SpaTe makes use of syntactic information provided in the dependency parse as well as information provided by SUTime and the NER system. Details as well as a discussion on the precision of the annotations can be found in Appendix C in [Gashteovski et al., 2019].

We subsequently refer to a triple with any spatial or temporal annotation as a *spatial/temporal triple*. MinIE-SpaTe differentiates between three types of such spatial or tem-

poral annotations: (i) annotations on entire triples, (ii) annotations on arguments, and (iii) spatial or temporal references. In what follows, we briefly discuss these types for temporal annotations; similar distinctions apply to spatial annotations.

Temporal annotations on triples provide temporal context for the entire triple. For example, from the sentence *"Bill Gates founded Microsoft in 1975."*, MinIE–SpaTe extracts triple *("Bill Gates"; "founded"; "Microsoft")* with temporal annotation ("in", 1975). Here, "in" is a *lexicalized temporal predicate* and 1975 is the *core temporal expression*. MinIE-SpaTe outputs *temporal modifiers* when found; e.g. the temporal annotation for *"... at precisely 11:59 PM"* is *("at", "11:59 PM", premod: precisely))*. We use the TIMEX3 format [Saurı et al., 2006] for representing the temporal information about the triple.

Sometimes the arguments of a triple contain some temporal information that refers to a phrase but not to the whole triple. MinIE provides temporal annotations for arguments for this purpose. For example, from the sentence *"Isabella II opened the 17th-century Parque del Retiro."*, MinIE-SpaTe extracts *("Isabella II"; "opened"; "Parque del Retiro")* with a temporal annotation (17th-century, Parque del Retiro) for the object argument. Generally, the temporal annotation contains information on its target (e.g., object), the temporal expression (17th-Century) and the head word being modified by the temporal expression (Retiro).

Finally, some triples contain temporal references as subject or object; MinIE-SpaTe annotates such references. For example, from the input sentence *"2003 was a hot year."*, MinIE-SpaTe extracts the triple: *("2003"; "was"; "hot year")*, where the subject (*2003*) is annotated with a temporal reference.

### 3.4 Postprocessing

During postprocessing, we remove clearly wrong triples, annotate the remaining triples with a confidence score, and ensure that Wikipedia links are not broken up.

In particular, MinIE retains the NER types provided during preprocessing in its extractions. We aggregated the most frequent relations per argument type and found that many of the triples were of the form *(person; "be"; organization)*, *(location; "be"; organization)*, and so on. These extractions almost always stemmed from an incorrect dependency parses obtained from sentences containing certain conjunctions. We filtered all triples with lemmatized relation *"be"* and different NER types for subject and object from MinIE-SpaTe's output.

In order to estimate whether or not a triple is correctly extracted, we followed ReVerb [Fader et al., 2011] and trained a logistic regression classifier. We used the labeled datasets provided by Gashteovski et al. [2017] as training data. Features were constructed based on an in-depth error analysis. The most important features were selected using chi-square relevance tests and include features such as the clause type, whether a coordinated conjunction has been processed, and whether or not MinIE simplified the triple. See Appendix D in [Gashteovski et al., 2019] for a complete list of features, and Sec. 4.6 for a discussion of confidence scores and precision in our corpora.

In a final postprocessing step, we rearranged triples such that links within the triples are not split across its constituents. For example, the triple *("Peter Brooke"; "was member of"; "Parliament")* produced by MinIE splits up the linked phrase "member of Parliament",

which results an incorrect links for the object (since it does not link to *Parliament*, but to *member of parliament*). We thus rewrite such triples to *("Peter Brooke"; "was"; "member of Parliament")*.

### 3.5 Provided Metadata

All metadata collected in the pipeline are retained for each triple, including provenance information, syntactic annotations, semantic annotations, and confidence scores. A full description of the provided metadata fields can be found in Appendix B in [Gashteovski et al., 2019].

### 3.6 Filtering

We constructed the OPIEC-Clean and OPIEC-Linked subcorpus by filtering the OPIEC corpus. OPIEC-Clean generally only retains triples between entities or concepts, whereas OPIEC-Linked only retains triples in which both arguments are linked. The filtering rules are described in more details in the following section.

## 4. Statistics

Basic statistics such as corpus sizes, frequency of various semantic annotations, and information about the length of the extracted triples of OPIEC and its subcorpora are shown in Tab. 2. We first discuss properties of the OPIEC corpus, then describe how we constructed the OPIEC-Clean and OPIEC-Linked subcorpora, and finally provide more in-depth statistics.

### 4.1 The OPIEC Corpus

The OPIEC corpus contains all extractions produced by MinIE-SpaTe. We analyzed these extractions and found that a substantial part of the triples are more difficult to handle by downstream applications. We briefly summarize the most prevalent cases of such triples; all these triples are filtered out in OPIEC-Clean.

First of all, a large part of the triples are under-specific in that additional context information from the extraction source is required to obtain a coherent piece of information. By far the main reason for under-specificity is lack of coreference information. In particular, 22% of the arguments in OPIEC are personal pronouns, such as in the triple *("He"; "founded"; "Microsoft")*. Such triples are under-specific because provenance information is needed to resolve what "He" refers to. Similarly, about 1% of the triples have determiners as arguments (e.g. *("This"; "lead to"; "controversy")*), and 0.2% Wh-pronouns (e.g. *("what"; "are known as"; "altered states of consciousness")*). Coreference resolution in itself is a difficult problem, but the large fraction of such triples shows that coreference resolution is important to further boost the recall of OIE systems.

Another problem for OIE systems are entity mentions—most notably for works of art—that constitute clauses. For example, the musical *"Zip Goes a Million"* may be interpreted as a clause, leading to the incorrect extraction *("Zip"; "Goes"; "a Million")*. A preliminary study showed that almost 30% of all the OPIEC triples containing the same recognized

|                                        | OPIEC          | OPIEC-Clean    |        | OPIEC-Linked   |       |
|----------------------------------------|----------------|----------------|--------|----------------|-------|
| Total triples (millions)               | 341.0          | 104.0          |        | 5.8            |       |
| Triples with semantic annotations      | 166.3 (49%)    | 51.46          | (49%)  | 3.37           | (58%) |
|    negative polarity    | 5.3   (2%)     | 1.33           | (1%)   | 0.01           | (0%)  |
|    possibility modality | 13.9   (4%)    | 3.27           | (3%)   | 0.04           | (1%)  |
|    quantities           | 59.4 (17%)     | 15.91          | (15%)  | 0.45           | (8%)  |
|    attribution          | 6.4   (2%)     | 1.44           | (1%)   | 0.01           | (0%)  |
|    time                 | 65.3 (19%)     | 19.66          | (19%)  | 0.58           | (1%)  |
|    space                | 61.5 (18%)     | 22.11          | (21%)  | 2.64           | (45%) |
|    space OR time        | 111.3 (33%)    | 37.22          | (36%)  | 3.01           | (52%) |
|    space AND time       | 15.4   (5%)    | 4.54           | (4%)   | 0.20           | (4%)  |
| Triple length in tokens ($\mu \pm \sigma$)       | $7.66 \pm 4.25$ | $6.06 \pm 2.82$ | | $6.45 \pm 2.65$ | |
|    subject ($\mu \pm \sigma$)     | $2.12 \pm 2.12$ | $1.48 \pm 0.79$ | | $1.92 \pm 0.94$ | |
|    relation ($\mu \pm \sigma$)    | $3.01 \pm 2.47$ | $3.10 \pm 2.56$ | | $2.77 \pm 2.14$ | |
|    object ($\mu \pm \sigma$)      | $2.52 \pm 2.69$ | $1.48 \pm 0.79$ | | $1.76 \pm 0.94$ | |
| Confidence score ($\mu \pm \sigma$)    | $0.53 \pm 0.23$ | $0.59 \pm 0.23$ | | $0.61 \pm 0.26$ | |

Table 2: Statistics for different OPIEC corpora. All frequencies are in millions. We count triples with annotations (not annotations directly). Percentages refer to the respective subcorpus.

named entity in both subject and object were of such a type. These cases constitute around 1% of OPIEC.

Finally, a substantial fraction of the triples in OPIEC has complicated expressions in its arguments. Consider for example the sentence *"John Smith learned a great deal of details about the U.S. Constitution."*. MinIE extracts the triple *("John Smith"; "learned"; "great deal of details about U.S. Constitution")*, which has a complicated object and is thus difficult to handle. A minimized variant such as *("John Smith"; "learned about"; "U.S. Constitution")* looses some information, but it expresses the main intent in a simpler way.

The above difficulties make the OPIEC corpus a helpful resource for research on improving or reasoning with complex OIE extractions rather than for downstream tasks.

### 4.2 The OPIEC-Clean Corpus

The OPIEC-Clean corpus is obtained from OPIEC by simply removing underspecified and complex triples. In particular, we consider a triple *clean* if the following conditions are met: (i) each argument is either linked, an entity recognized by the NER tagger, or matches a Wikipedia page title, (ii) links or recognized named entities are not split up across constituents, and (iii) the triple has a non-empty object.

Conditions (i) and (ii) rule out the complex cases mentioned in the previous section. Note that we ignore quantities (but no other modifiers) when checking condition (i). For example, the triple *("$Q_1$ electric locomotives"; "were ordered from"; "Alsthom")* with $Q_1$ = *"Three"* is considered clean; here *"electric locomotives"* holds a link to *TCDD E4000* and *"Alsthom"* holds a link to *Alsthom*.

MinIE is a clause-based OIE system and can produce extractions for so-called SV clauses: these extractions consists of only a subject and a relation, but no object. 3.5% of the triples in OPIEC are of such type. An example is the triple *("Civil War"; "have escalated"; "")*. Although such extractions may contain useful information, we exclude them via condition (iii) to make the OPIEC-Clean corpus uniform.

Roughly 30% of the triples (104M) in OPIEC are clean according to the above constraints. Table 2 shows that clean triples are generally shorter on average and tend to have a higher confidence score than the full set of triples in OPIEC. The OPIEC-Clean corpus is easier to work with than the full OPIEC corpus; it is targeted towards both downstream applications and research in automated knowledge base construction.

### 4.3 The OPIEC-Linked Corpus

The OPIEC-Linked corpus contains only those triples from OPIEC-Clean in which both arguments are linked. Although the corpus is much smaller than OPIEC-Clean (5.8M triples, i.e., roughly 5.5% of OPIEC-Clean), it is the largest corpus to date with golden disambiguation links for the arguments. We use the corpus mainly to compare OIE extractions with the information present in the DBpedia and YAGO knowledge bases; see Sec. 5.

### 4.4 Semantic Annotations

About 49% of all triples in OPIEC contain some sort of semantic annotation (cf. Tab. 2); in OPIEC-Linked, the fraction increases to 58%. Most of the semantic annotations referred to quantities, space or time; these annotations provide important context for the extractions. There is a significantly smaller amount of negative polarity and possibility modality annotations. One reason for the lack of such annotations may be in the nature of the Wikipedia articles, which aim to contain encyclopedic, factual statements and are thus more rarely negated or hedged.

The distribution of annotations is similar for OPIEC and OPIEC-Clean, but differs significantly for OPIEC-Linked. In particular, we observe a drop in quantity annotations in OPIEC-Linked because most of the linked phrases do not contain quantities. The fraction of spatial triples in OPIEC-Linked is also much higher than the rest of the corpora. The reason is because Wikipedia contains many pages for locations, which are also linked and contain factual knowledge, thus resulting in many triples with spatial reference (e.g. the triple *("Camborne School of Mines"; "was relocated to"; "Penryn")*).

### 4.5 NER Types and Frequent Relations

For OPIEC-Clean, Fig. 2 shows the fraction of arguments and argument pairs that are recognized as named entities by the NER tagger, along with the NER type distribution of the arguments.

Out of the around 208M arguments, roughly 42% are recognized named entities. The most frequent NER type is *person*, followed by *location* and *organization*. The remaining NER types are not that frequent (less than 3% each). On the other hand, 58% of the arguments are not typed. These are mostly concepts (more precisely, strings that match Wikipedia pages not referring to an entity) and are thus not recognized by the NER system.

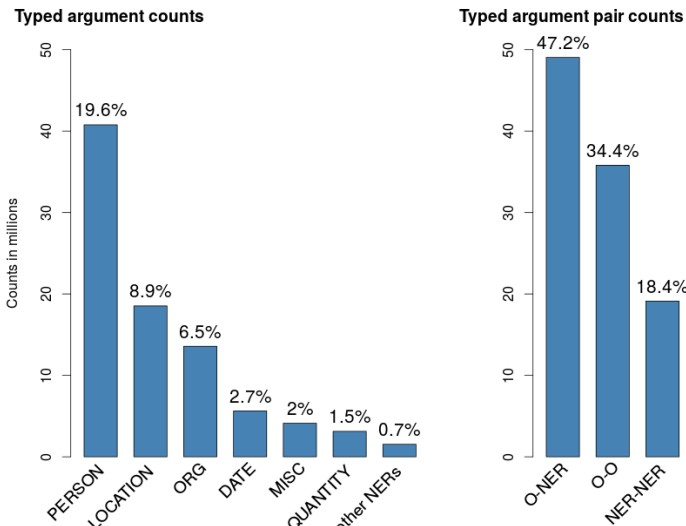

Figure 2: Distribution of NER types for arguments and argument pairs in OPIEC-Clean. Here "O" refers to arguments that are not recognized as a named entity.

The top-10 most frequent arguments which are not typed are the words *film, population, city, village, father, song, company, town, album* and *time*, with frequencies varying between 408k and 616k.

Fig. 2 also reports the fraction of triples in which none, one, or both arguments are recognized as a named entity. We found that 18% of the triples (19M) in OPIEC-Clean have two typed arguments, and around 66% of the triples (68M) have at least one typed argument. Thus the majority of the triples involves named entities. 34% of the triples do not have recognized named entity arguments.

Tab. 3 shows the most frequent open relations between arguments recognized as persons and/or locations (which in turn are the top-3 most frequent argument type pairs). We will analyze some of the open relations in more detail in Sec. 5. For now, note that the most frequent relation between persons is *"have"*, which is highly polysemous. Other relations, such as *"marry"* and *"be son of"*, are much less ambiguous. We provide all open relations between recognized argument types as well as their frequencies with the OPIEC-Clean corpus.

### 4.6 Precision and Confidence Score

Each triple in the OPIEC corpora is annotated with a confidence score for correct extraction. To evaluate the accuracy of the confidence score, we took an independent random sample of triples (500 in total) from OPIEC and manually evaluated the correctness of the triples following the procedure of Gashteovski et al. [2017].[3] We found that 355 from the 500 triples (71%) were correctly extracted. Next, we bucketized the triples by confidence score into ten equi-width intervals and calculated the precision within each interval; see Fig. 3a.

---

3. The annotation guidelines are provided at http://www.uni-mannheim.de/media/Einrichtungen/dws/Files_Research/Software/MinIE/minie-labeling-guide.pdf

| PERSON-PERSON | | LOCATION-LOCATION | | PERSON-LOCATION | |
|---|---|---|---|---|---|
| "have" | (130,019) | "be in" | (2,126,562) | "be bear in" | (203,091) |
| "marry" | (49,405) | "have" | (40,298) | "die in" | (37,952) |
| "be son of" | (40,265) | "be village in adminis-trative district of" | (9,130) | "return to" | (36,702) |
| "be daughter of" | (37,089) | "be north of" | (3,816) | "move to" | (36,072) |
| "be bear to" | (29,043) | "be suburb of" | (3,291) | "be in" | (25,847) |
| "be know as" | (25,607) | "be west of" | (3,238) | "live in" | (22,399) |
| "defeat" | (22,151) | "be part of" | (3,188) | "grow up in" | (17,571) |

Table 3: Most frequent open relations between persons and locations (as recognized by the NER tagger) in OPIEC-Clean

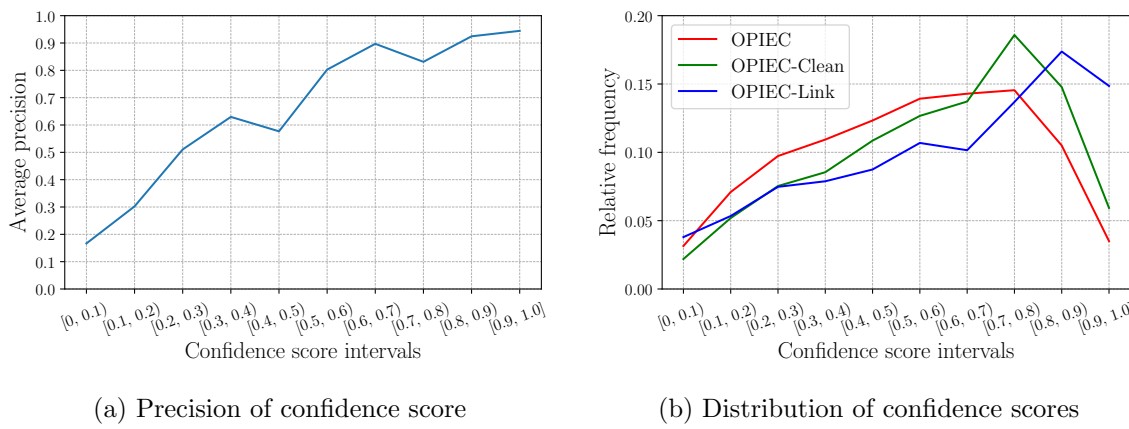

(a) Precision of confidence score

(b) Distribution of confidence scores

Figure 3: Precision and distribution of confidence scores

We found that the confidence score is highly correlated to precision (Pearson correlation of $r = 0.95$) and thus provides useful information.

The distribution in confidence scores across the various corpora is shown in Fig. 3b. We found that OPIEC-Clean and, in particular, OPIEC-Link contain a larger fraction of high-confidence triples than the raw OPIEC corpus: these corpora are cleaner in that more potentially erroneous extractions are filtered out. Although triples with lower confidence score tend to be more inaccurate, they may still provide suitable information. We thus included these triples into our corpora; the confidence scores allow downstream applications to handle these triples as appropriate.

## 5. Analysis

In this section, we compare the information present in the OIE triples in OPIEC with the information present in the DBpedia [Auer et al., 2007] and YAGO [Hoffart et al., 2013] knowledge bases. Since all resources extract information from Wikipedia— OPIEC from the text and DBpedia as well as YAGO from semi-structured parts of Wikipedia—, we wanted to understand whether and to what extent they are complementary. Generally, the

| location | | associatedMusicalArtist | | spouse | |
|---|---|---|---|---|---|
| *"be in"* | (43,842) | *"be"* | (6,273) | *"be wife of"* | (1,965) |
| *"have"* | (3,175) | *"have"* | (3,600) | *"be"* | (1,308) |
| *"be"* | (1,901) | *"be member of"* | (740) | *"marry"* | (702) |
| *"be at"* | (1,109) | *"be guitarist of"* | (703) | *"be widow of"* | (479) |
| *"be of"* | (706) | *"be drummer of"* | (458) | *"have"* | (298) |
| *"be historic home located at"* | (491) | *"be feature"* | (416) | *"be husband of"* | (284) |

Table 4: The most frequent open relations aligned to the DBpedia relations *location*, *associatedMusicalArtist*, and *spouse* in OPEIC-Linked

disambiguation of OIE triples w.r.t. a given knowledge base is in itself a difficult problem. We avoid this problem here by (1) restricting ourselves to the OPIEC-Linked corpus (for which we have golden entity links) and (2) focusing on statistics that do not require a full disambiguation of the open relations but are nevertheless insightful.

### 5.1 Alignment With Knowledge Bases

To align the OIE tripes from OPIEC-Linked to YAGO or DBpedia, we make use of the distant supervision assumption. For each open triple $(s, r_{\text{open}}, o)$ from OPIEC-Linked, we search the KB for any triple of form $(s, r_{\text{KB}}, o)$ or $(o, r_{\text{KB}}, s)$. Here $s$ and $o$ refer to disambiguated entities, whereas $r_{\text{open}}$ refers to an open relation and $r_{\text{KB}}$ to a KB relation. If such a triple exists, we say that triple $(s, r_{\text{open}}, o)$ has a *KB hit*, and that $r_{\text{open}}$ *is aligned with* $r_{\text{KB}}$. In fact, if $r_{\text{open}}$ (e.g., *"was born in"*) is a mention of KB relation $r_{\text{KB}}$ (e.g., *birthPlace*), and subject and object are not swapped, then both triples express the same information. Otherwise, the open triple and the KB triple express different information (e.g., *deathPlace*) or inverse relations (e.g., *isBirthplaceOf*). We can thus think of the number of KB hits as an optimistic measure of the number of open triples that are represented in the KB (with caveats, see below): the KB contains some relation between the corresponding entities, although not necessarily the one being mentioned.

We observed that 29.7% of the OIE triples in OPIEC-Linked have a KB hit in either DBpedia or YAGO. More specifically, 25.5% of the triples have a KB hit in DBpedia, 20.8% in YAGO, and 16.6% in both DBpedia and YAGO. Most of these triples have exactly one hit in the corresponding KB. Consequently, 70.3% of the linked triples do not have a KB hit; we analyze these triples below.

Tab. 4 shows the most frequent open relations aligned to the DBpedia relations *location*, *associatedMusicalArtist*, and *spouse*. The frequencies correspond to the number of OIE triples that (1) have the specified open relation (e.g., *"be wife of"*) and (2) have a KB hit with the specified KB relation (e.g., *spouse*). There is clearly no 1:1 correspondence between open relations and KB relations. On the one hand, open relations can be highly ambiguous (e.g., *"be"* has hits to *location* and *associatedMusicalArtits*). On the other hand, open relations can also be more specific than KB relations (e.g., *"be guitarist of"* is more specific than *associatedMusicalArtist*) or semantically different (e.g., *"be widow of"* and *spouse*) than the KB relations they align to.

To gain more insight into the type of triples contained in OPIEC-Clean, we selected the top-100 most frequent open relations for further analysis. These relations constitute roughly 38% of the OPIEC-Clean corpus, relation frequencies are thus highly skewed. We then used OPIEC-Linked as a proxy for the number of DBpedia hits of these relations. The results are summarized in Tab. 5 as well as in Appendix A in [Gashteovski et al., 2019]. The open relation *"have"*, for example, is aligned to 330 distinct DBpedia relations, the most frequent ones being *author, director,* and *writer.* Generally, the fraction of KB hits (from OPIEC-Linked) is quite low, averaging at 16.8% for the top-100 relations. This indicates that there is a substantial amount of information present in OIE triples that is not present in KBs. Moreover, about 42 distinct KB relations align on average with each open relation, which again indicates that open relations should not be directly mapped to KB relations.

By far the most frequent open relations in OPIEC-Clean are *"be"* and *"have"*, which constitute 21.1% and 6.1% of all the triples, respectively. These open relations are also the most ambiguous ones in that they are aligned with 410 and 330 different DBpedia relations, respectively. Here the open relations are far more "generic" than the KB relations that they are aligned to. This is illustrated in the following examples:

*("Claudia Hiersche"; "be"; "actress")* $\xrightarrow{DBpedia}$ (Claudia_Hiersche; occupation; Actress)

*("Cole Porter"; "have"; "Can-Can" )* $\xrightarrow{DBpedia}$ (Can-Can_(musical); musicBy; Cole_Porter)

$\xrightarrow{DBpedia}$ (Can-Can_(musical); lyrics; Cole_Porter)

*("Oddbins"; "have"; "Wine")* $\xrightarrow{DBpedia}$ (Oddbins; product; Wine)

Note that in these cases, "have" refers to the possessive (e.g., Odbbins' Wine).

### 5.2 Spatio-Temporal Facts

We also investigated to what extent the space and time annotations in OIE triples relate to corresponding space and time annotations in YAGO. In particular, YAGO provides:

- *YAGO date facts*, which have entities as subjects and dates as objects: e.g., (Keith_Joseph, wasBornOnDate, 1918-01-17).

- *YAGO meta-facts*, which are spatial or temporal information about other YAGO facts: e.g., (Steven_Lennon, playsFor, Sandnes_Ulf) has meta-fact (occursUntil, 2014).

Note that date facts roughly correspond to temporal reference annotations in OPIEC, wheres meta-facts correspond to spatial or temporal triple annotations.

To compare OPIEC with YAGO date facts, we selected all triples with (i) a disambiguated subject and (ii) an object that is annotated as date from OPIEC. There are 645,525 such triples. As before, we align these triples to YAGO using an optimistic notion of a KB hit. In particular, a *KB date hit* for an OIE date fact $(s, r_{open}, d_{open})$ is any KB fact of form $(s, r_{KB}, d_{KB})$, i.e., we require that there is temporal information but ignore whether or not it matches. We use this optimistic notion of KB date hit to avoid disambiguating the open relation or date. Even with this optimistic notion, we observed that only 36,262 (5.6%) of the OIE date facts have a KB date hit in the YAGO date facts.

We also compared the spatial and temporal annotations of OPIEC-Linked with the YAGO meta-facts. We found that roughly 13,203 OPIEC-Linked triples have a KB hit

| Open relation | Frequency in OPIEC-Clean | Frequency in OPIEC-Link | # KB hits | # distinct KB rel.s | Top-3 aligned DBpedia rel. and hit frequency | |
|---|---|---|---|---|---|---|
| "be" | 21,911,174 | 1,475,332 | 173,107 (11.7%) | 410 | type
occupation
isPartOf | 72,077
12,508
8,012 |
| "have" | 6,369,086 | 216,332 | 137,865 (63.7%) | 330 | author
director
writer | 14,056
10,416
9,765 |
| "be in" | 3,219,301 | 1,150,667 | 804,378 (69.9%) | 225 | country
isPartOf
state | 287,557
222,175
64,675 |
| "include" | 487,899 | 14,746 | 1,573 (10.7%) | 128 | type
associatedBand
associatedMusicalArtist | 380
83
83 |
| "be bear in" | 289,947 | 7,138 | 1,477 (20.7%) | 30 | birthPlace
isPartOf
deathPlace | 1,147
73
62 |
| "win" | 236,169 | 8,819 | 910 (10.3%) | 54 | award
race
team | 299
210
50 |
| "be know as" | 215,809 | 7,993 | 675 (8.4%) | 123 | location
associatedBand
associatedMusicalArtist | 46
42
42 |
| "become" | 213,807 | 5,123 | 393 (7.7%) | 90 | successor
associatedBand
associatedMusicalArtist | 63
33
33 |
| "have be" | 191,140 | 1,855 | 101 (5.4%) | 32 | type
position
leader | 12
9
7 |
| "play" | 163,643 | 4,842 | 835 (17.2%) | 54 | portrayer
author
instrument | 367
101
76 |
| "be know" | 157,751 | 351 | 51 (14.5%) | 15 | occupation
family
country | 20
12
3 |
| "die in" | 146,681 | 638 | 127 (19.9%) | 20 | deathPlace
battle
commander | 71
13
9 |
| "join" | 134,159 | 2,656 | 903 (34.0%) | 65 | team
associatedBand
associatedMusicalArtist | 301
105
105 |

Table 5: The most frequent open relations in OPIEC-Clean, along with DBpedia alignment information from OPIEC-Link (continued in Tab. 6, Appendix A in [Gashteovski et al., 2019])

with a YAGO triple that also has an associated a meta-fact. Out of these linked OIE triples, 2,613 are temporal and 2,629 are spatial.

To provide further insight, we analyzed the spatial-temporal annotations of OPIEC more closely. We identified two major reasons why spatio-temporal information is not found in KBs: (i) the information in missing from the KB, and (ii) the information is available only indirectly. For an example of missing information, consider the OPIEC-Linked triple *("Iain Duncan Smith", "is leader of", "Conservative Party")* with temporal annotations *(pred="from", 2001)* and *(pred="to", 2003)*. YAGO contains the KB hit (Iain_Duncan_Smith; isAffiliatedTo; Conservative_Party_(UK)). Note that the YAGO relation is less specific than the open relation, and that no temporal information is present. As another example, consider the OIE triple *("Neue Nationalgalerie"; "be built by"; "Ludwig Mises van der Rohe")* with temporal annotation *(pred="in", 1968)*. Again, the YAGO hit (Neue_Nationalgalerie; linksTo; Ludwig_Mies_van_der_Rohe) is less specific and lacks temporal information. YAGO does contain the triple (Neue_Nationalgalerie; hasLongitude; 13.37) with temporal meta-fact 1968-01-01. Here the temporal information is present in the KB, but only indirectly and for a different relation.

Generally, the low number of KB hits indicates that a wealth of additional spatial and/or temporal information is present in OIE data, and that the spatial/temporal annotations provided in OPIEC are potentially very valuable for automated KB completion tasks.

### 5.3 Non-Aligned OIE Triples

We found that more than half of the triples in OPIEC-Linked that do not have a KB hit refer to one of the top-100 most frequent relations in OPIEC-Clean. Since OPIEC-Clean is much larger than OPIEC-Linked, this indicates that it contains many facts not present in DBpedia. Naturally, not all these facts are correct, though, and disambiguation is a major challenge. In particular, we took a random sample of 100 non-aligned triples from OPIEC-Linked and manually labeled each triple as *correctly extracted* or *incorrectly extracted*. 60% of the triples were considered to be correctly extracted. In another sample of 100 high-confidence triples (score > 0.5), 80% were correctly extracted. This shows the potential and the challenges of harnessing the knowledge contained within the OIE triples.

### 6. Conclusions

We created OPIEC, a large open information extraction corpus extracted from Wikipedia. OPIEC consists of hundreds of millions of triples, along with rich metadata such as provenance information, syntactic annotations, semantic annotations, and confidence scores. We reported on a data profiling study of the OPIEC corpus as well as subcorpora. In particular, we analyzed to what extent OPIEC overlaps with the DBpedia and YAGO knowledge bases. Our study indicates that most open facts do not have counterparts in the KB such that OIE corpora contain complementary information. For the information that overlaps, open relation are often more specific, more generic, or simply correlated to KB relations (instead of semantically equivalent). We hope that the OPIEC corpus, its subcorpora, derived statistics, as well as the codebase used to create the corpus are a valuable resource for automated KB construction and downstream applications (for example, an independent study showed the utility of OPIEC in entity-aspect linking [Nanni et al., 2019]).

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
