# OpenReview forum: "OPIEC: An Open Information Extraction Corpus"
_AKBC.ws/2019/Conference — AKBC 2019_

### Official Review · AnonReviewer1 · 2018-12-18
**Interesting analysis, but may not be enough content**

**Rating:** 6
**Confidence:** 4

**Review:**

The paper describes the creation of OPIEC -- an Open IE corpus over English Wikipedia.
The corpus is created in a completely automatic manner, by running an off-the-shelf OIE system (MinIE), which yields 341M SVO tuples. Following, this resource is automatically filtered to identify triples over named entities (using an automatic NER system, yielding a corpus of 104M tuples), and only entities which match entries in Wikipedia (5.8M tuples).

On the positive side, I think that resources for Open IE are useful, and can help spur more research and analyses.

On the other hand, however, I worry that OPIEC may be too skewed towards the predictions of a specific OIE system, and that the work presented here consists mainly of running off-the-shelf can be extended to contain more novel substance, such as a new Open IE system and its evaluation against this corpus, or more dedicated manual analysis. For example, I believe that most similar resources (e.g., ReVerb tuples) were created as a part of a larger research effort.

The crux of the matter here I think, is the accuracy of the dataset, reported tersely in Section 5.3, in which a manual analysis (who annotated? what were their guidelines? what was their agreement?) finds that the dataset is estimated to have 60% correct tuples. Can this be improved? Somehow automatically verified?

Detailed comments:

- I think that the paper should make it clear in the title or at least in the abstract that the corpus is created automatically by running an OIE system on a large scale. From current title and abstract I was wrongfully expecting a gold human-annotated dataset.

- Following on previous points, I think the paper misses a discussion on gold vs. predicted datasets for OIE, and their different uses. Some missing gold OIE references:
Wu and Weld (2010),  Akbik and Loser (2012), Stanovsky and Dagan (2016).

- Following this line, I don't think I agree with the claim in Section 4.3 that "it is the largest corpus with golden annotations to date". As far as I understand, the presented corpus is created in a completely automated manner and bound to contain prediction errors.

- I think that some of the implementation decisions seem sometimes a little arbitrary. For instance, for the post-processing example which modifies
(Peter Brooke; was a member of; Parliament) to (Peter Brooke; was ; a member of Parliament), I think I would've preferred the original relation, imagining a scenario where you look for all members of parliament (X; was a member of; Parliament), or all of the things Peter Brooke was a member of (Peter Brooke; was a member of; Y) seems more convenient to me.

Minor comments & typos:

- I assume that in Table 1, unique relations and arguments are also in millions? I think this could be clearer, if that's the indeed the case.

- I think it'd be nice to add dataset sizes to each of the OPIEC variants in Fig 1.

- End of Section 3.1 "To avoid loosing this relationship" -> "losing this relationship"

- Top of P. 6: "what follows, we [describe] briefly discuss these"

- Section 4.5 (bottom of p. 9) "NET type" -> "NER type"?

---

> ### Author Response · Authors · 2019-01-31
> **Clarifications**
>
> "I worry that OPIEC may be too skewed towards the predictions of a specific OIE system"
>
> -> Yes, OPIEC is based on a modified variant of MinIE, which allows us to provide syntactic annotations, semantic annotations, and confidence scores. The pipeline used to create OPIEC can be applied to other datasets and (perhaps with minor modifications) with other OIE systems as well. We plan to publish the source code of this pipeline along with the corpus and data access tools along with OPIEC.
>
>
> "the work presented here consists mainly of running off-the-shelf can be extended to contain more novel substance, such as a new Open IE system and its evaluation against this corpus".
>
> -> We use an improved version of MinIE, which adds space-time awareness + confidence score. MinIE, and in particular the OPIEC corpus, is indeed being used in other research projects already. The goal of this paper is to introduce the dataset to other researchers and provide insight into its properties.
>
>
> "The crux of the matter here I think, is the accuracy of the dataset, reported tersely in Section 5.3, in which a manual analysis (who annotated? what were their guidelines? what was their agreement?) finds that the dataset is estimated to have 60% correct tuples. Can this be improved? Somehow automatically verified?"
>
> -> We added subsection 4.6 to provide more information about the confidence scores that OPIEC provides. These scores allow filtering the corpus for only high-confidence triples, for example.
>
>
> "I think that the paper should make it clear in the title or at least in the abstract that the corpus is created automatically by running an OIE system on a large scale. From current title and abstract I was wrongfully expecting a gold human-annotated dataset."
> "I don't think I agree with the claim in Section 4.3 that "it is the largest corpus with golden annotations to date". As far as I understand, the presented corpus is created in a completely automated manner and bound to contain prediction errors."
>
> -> We carefully revisited the paper to be more precise. For example, we now always refer to “golden annotations for arguments”.
>
>
> "Following on previous points, I think the paper misses a discussion on gold vs. predicted datasets for OIE, and their different uses. Some missing gold OIE references: Wu and Weld (2010),  Akbik and Loser (2012), Stanovsky and Dagan (2016)."
>
> -> To the best of our knowledge, no data was released from Wu and Weld (2010) and Akbik and Loser (2012). We now mention the benchmark dataset of Stanovsky and Dagan (2016) in the related work section. The focus of this work is on large OIE resources, however.
>
>
> - "I think that some of the implementation decisions seem sometimes a little arbitrary. For instance, for the post-processing example which modifies (Peter Brooke; was a member of; Parliament) to (Peter Brooke; was ; a member of Parliament), I think I would've preferred the original relation, imagining a scenario where you look for all members of parliament (X; was a member of; Parliament), or all of the things Peter Brooke was a member of (Peter Brooke; was a member of; Y) seems more convenient to me."
>
> -> We agree with the reviewer in this particular example. However, “Member of Parliament” is a concept in Wikipedia. OPIEC avoids to split concepts or named entities across arguments/relations; such an approach is often erroneous.
>
>
> “I assume that in Table 1, unique relations and arguments are also in millions? I think this could be clearer, if that's the indeed the case.”
>
> -> Fixed
>
> - I think it'd be nice to add dataset sizes to each of the OPIEC variants in Fig 1.
> -> Fixed
>
> - Typos -> fixed

---

### Official Review · AnonReviewer3 · 2018-12-21
**Good paper on producing a triple store from Wikipedia articles.**

**Rating:** 7
**Confidence:** 4

**Review:**

This paper presents a dataset of open-IE triples that were collected from Wikipedia with the help of a recent extraction system.

This venue seems like an ideal fit for this paper and I think it would make a good addition to the conference program. While there is little technical originality, the overall execution of the experimental part is quite good and I like that the authors focused in their report on describing how they filtered the output of the employed IE system and that they present interesting examples from the conducted filtering steps.

I particularly liked the section on comparing the new resource to the existing knowledge bases from the same source (YAGO, DBpedia), I think it makes a lot of sense to pick resources that leverage other parts of Wikipedia (category system, ...) and not the main article text, and to look into how coverage of the these different approaches relates.

It would have been nice to also compare against other datasets/triple stores/... that used open-IE to extract from Wikipedia. A couple of references discussing such are listed in the paper, e.g., DefIE or KB-Unify seem like good candidates.

---

### Official Review · AnonReviewer2 · 2019-01-09
**OPIEC: An Open Information Extraction Corpus**

**Rating:** 7
**Confidence:** 3

**Review:**

In this paper, the authors build a new corpus for information extraction which is larger comparing to the prior public corpora and contains information not existing in current corpora. The dataset can be useful in other applications. The paper is well written and easy to follow. It also provides details about the corpus. However, there are some questions for the authors: 1) It uses the NLP pipeline and the MinIE-SpaTe system. When you get the results, do you evaluate to what extent that the results are correct? 2) In Section 3.4, the author mentioned the correctness is around 65%, what do you do for those incorrect tuples? 3) Have you tried any task-based evaluation on your dataset?

---

> ### Author Response · Authors · 2019-01-31
> **Clarifications**
>
> "1) It uses the NLP pipeline and the MinIE-SpaTe system. When you get the results, do you evaluate to what extent that the results are correct?"
> "2) In Section 3.4, the author mentioned the correctness is around 65%, what do you do for those incorrect tuples?"
>
> -> We report on precision and confidence scores in the new subsection 4.6.
>
>
> "3) Have you tried any task-based evaluation on your dataset?"
>
> The OPIEC corpus is being used in other research projects. The goal of this paper is to introduce the dataset to other researchers and provide insight into its properties.

---

### Author Response · Authors · 2019-01-31
**New subsection added addressing common reviewers comments**

Thank you very much for your helpful and insightful comments! All of you asked for more information about precision; we added Section 4.6 “Precision and Confidence Score” to provide more details and statistics about precision as well as the confidence scores provided with the dataset.

---

### Meta-Review · Area_Chair1 · 2019-02-12
**A nice dataset paper**

**Recommendation:** Accept (Poster)
**Confidence:** 4

**Metareview:**

This paper describes a new Open IE corpus over English Wikipedia. All the reviewers agree this paper is suitable for this venue and the dataset is useful. Overall, the paper is well-written and the experiments are convincing. Despite the novelty of this paper is  relatively thin, it is a decent paper.

---

### Decision · Program_Chairs · 2019-02-15
**AKBC 2019 Conference Decision**

Accept